# The Origin of Life in the Early Continental Crust: A Comprehensive Model

**DOI:** 10.3390/life15030433

**Published:** 2025-03-10

**Authors:** Ulrich Schreiber

**Affiliations:** Faculty of Biology, University of Duisburg-Essen, 45141 Essen, Germany; ulrich.schreiber@uni-due.de

**Keywords:** origin of life, hydrothermal biochemistry, information storage, continental crust model, supercritical fluids, pre-LUCA stage

## Abstract

Continental rift zones on the early Earth provided essential conditions for the emergence of the first cells. These conditions included an abundant supply of raw materials, cyclic fluctuations in pressure and temperature over millions of years, and transitions of gases between supercritical and subcritical phases. While evidence supports vesicle formation and the chemical evolution of peptides, the mechanism by which information was stored remains unresolved. This study proposes a model illustrating how interactions among organic molecules may have enabled the encoding of amino acid sequences in RNA. The model highlights the interplay between three key molecular components: a proto-tRNA, the vesicle membrane, and short peptides. The vesicle membrane acted as a reservoir for hydrophobic amino acids and facilitated their attachment to proto-tRNA. As a single strand, proto-tRNA also served as proto-mRNA, enabling it to be read by charged tRNAs. By replicating this information and arranging RNA strands, the first functional peptides such as pore-forming proteins may have formed, thus improving the long-term stability of the vesicles. This model further outlines how these vesicles may have evolved into the earliest cells, with enzymes and larger RNA molecules giving rise to tRNA and ribosomal structures. Shearing forces may have facilitated the first cellular divisions, representing a pre-LUCA stage.

## 1. Introduction

The origin of life remains one of the greatest unsolved mysteries, with the precise environment for the emergence of the first cell yet to be conclusively identified. Fundamental questions persist: where did the building blocks of life form in sufficient abundance, and how did they concentrate into reactive mixtures [1]? What energy sources drove the critical chemical reactions for life, and how did these processes sustain life despite the constraints of increasing entropy [2,3]? How was the high thermodynamic barrier for peptide condensation in aqueous solutions—commonly referred to as the “water paradox”—overcome [4,5]? If life began on Earth’s surface, what roles did solar wind and ultraviolet radiation play [6]? Furthermore, what impact did meteorite bombardment, erosion, and sedimentation have on ponds that might have existed over millions of years, fostering the conditions necessary for life [7]? What were the prevailing temperatures on the early Earth’s surface, and how did they influence cloud formation and precipitation patterns—key factors for sustaining dry–wet cycles? How did vesicles evolve as precursors to cells, and how did complex molecular sequences, which could not have arisen by chance, come into existence [8,9,10]? What accounts for the handedness of chiral molecules in living systems [11,12]? Lastly, how did information storage originate during the earliest stages of life [13]? These questions have been extensively discussed from various perspectives in review articles and books in recent years [14,15,16].

Until recently, no widely accepted model described a realistic, experimentally testable environment for the origin of life, despite the necessity of such a model to advance our understanding of the peptide/RNA system.

Two primary models have dominated the literature: the alkaline hydrothermal vents model at lower temperatures [17] and the “warm little ponds” model, which incorporates both hydrothermal inputs and extra-terrestrial organic compounds [18,19,20]. (The initially favored black smoker model was not pursued further, as it involves extreme temperatures and high concentrations of metal compounds). However, both models face significant challenges. Modern white smokers, for instance, are often associated with calcareous sediments from biogenic reefs, conditions that are unlikely to have existed on the early Earth [7,21]. Similarly, the “warm little ponds” model faces several difficulties, including exposure to ultraviolet radiation and solar wind particles, rapid sediment or salt accumulation, and the potential destruction of ponds by meteorite impacts, which could lead to catastrophic flooding and erosion. Additionally, no reliable estimate exists for the volume of organic material delivered to such ponds by meteorites, as organic concentrations in meteorites are low, with weathering releasing only trace amounts over extended periods [22].

Recent studies on early continental crust environments and the origin of life propose a novel model that can be tested both in the laboratory and in nature [23,24,25]. This model centers on the evolution of the continental crust, which is thought to have formed soon after Earth’s surface cooled. By the time life is believed to have emerged [15], the continental crust may have already constituted over 20% of its current mass [26,27]. Tectonic stresses likely created fault zones in the early continental crust, facilitating the release of magma and gases to the surface. These fault zones may have remained active for tens to hundreds of millions of years. Volcanic regions would have exhibited a range of temperatures, with cooler zones influenced by artesian water flows, potentially giving rise to cold-water geysers driven by CO_2_ and N_2_ gases. Such conditions in cold-water geysers can be reproduced in high-pressure laboratory experiments [24,25]. Additionally, the chemical composition of fluids from these fault zones can be studied through fluid inclusions in hydrothermally formed fissure minerals [28,29]. This approach offers new insights into the conditions that may have fostered the origin of life.

## 2. Environment and Resources

Tectonic faults in the upper continental crust, filled with water and gases, presented a diverse range of pressure and temperature conditions (Figure 1) [23]. Processes such as Haber–Bosch and Fischer–Tropsch syntheses may have produced life’s building blocks from abiotic materials, utilizing various metallic catalysts naturally available in these environments. These conditions closely resemble the early atmospheric conditions studied by Peters et al. (2023), which involved similar gas partial pressures and iron compounds derived from volcanic ash and meteorites acting as catalysts [30]. Today, subsurface ecosystems influenced by geological degassing represent microbial hotspots within the deep biosphere [31]. In archaic hydrothermal quartz from Western Australia, a variety of organic molecules, including precursors of lipid molecules such as aldehydes, have been identified. Similar results were observed in subrecent calcites from the marginal zones of Quarternary volcanic calderas in the Eifel region [28,29].

Hennet et al. demonstrated that amino acids could form under hydrothermal conditions (150 °C, 10 atm, pH 7) with mineral catalysts and starting compounds like formaldehyde, ammonia, and cyanide [32]. Glycine was detected in the highest concentrations, with other amino acids (aspartic acid, serine, glutamic acid, alanine, and isoleucine) present at concentrations roughly two orders of magnitude lower. Marshall [33] extended this work by identifying 12 amino acids formed under hydrothermal conditions, including proline, leucine, lysine, valine, threonine, and phenylalanine. These results suggest that tectonic fault zones could have supplied at least 12 amino acids crucial for early peptide synthesis, although their concentrations likely varied. Furthermore, LaRowe and Regnier demonstrated the synthesis of five nucleobases (adenine, cytosine, guanine, thymine, and uracil) as well as ribose and deoxyribose under hydrothermal conditions typical of such environments [34].

Fluids in crustal faults often exhibit low pH (<3) due to high pressures where CO_2_ dominates the gas phase [35]. Sulfur- and nitrogen-based compounds, with H_2_S and NH_3_ as fundamental molecules that may have formed in the early geochemical environment of Earth, influence pH differently: while sulfur compounds tend to decrease pH, nitrogen compounds generally contribute to its increase. High pressure, temperature, and low pH promote the alteration of marginal rocks, releasing phosphates from apatite, a mineral abundant in igneous rocks. Cations from rock-forming minerals can act as catalysts for organic reactions or form new reactive surfaces, such as sulfide ore veins, carbonates, or arsenites on fault walls. Sulfide ores, often containing radioactive elements, may have contributed to RNA evolution and reaction networks [36]. RNA is most stable at a pH of 4 to 5, becoming increasingly unstable at higher pH values. This suggests that its origin may be linked to acidic environments such as hydrothermal vents, volcanic lakes, or comet ponds. Supporting this idea, Bernhardt and Tate (2012) propose that RNA is well suited for an early Earth with acidic conditions, as its phosphodiester bonds, as well as aminoacyl-(t)RNA and peptide bonds, exhibit increased stability at low pH [37].

Clay minerals lining fault surfaces could have served as templates for nucleotide linkage, a critical step in RNA formation [38,39]. Quartz, commonly found in fault fissures, is often accompanied by colloidal SiO_2_-rich fluids that, like quartz surfaces, exhibit catalytic properties [40]. Additionally, during the early history of Earth, stronger tidal forces from a closer-Moon-influenced crustal activity [41]. While piezoelectric effects in crustal rocks typically cancel out during seismic events due to random mineral distribution, they can create regular voltage gradients within hydrothermally formed quartz in faults. These gradients could have generated weak cyclical electrical currents, and voltage spikes associated with seismic activity may have caused lightning-like discharges in gas-filled cavities or facilitated the splitting of water into H_2_ and O_2_ [42].

The stable conditions deep within Earth’s crust, shielded from solar wind, UV radiation, surface erosion, and flooding, could have persisted for millions of years. Organic molecules from surface environments may have been transported to these depths via artesian water flow and geyser backflow, potentially delivering meteoritic organic material. CO_2_ and N_2_ gases in geysers played a crucial role in this context. The upward flow of gas bubbles facilitated substance transport, continuously replenishing reactants and removing reaction products. This dynamic prevented equilibrium conditions and reduced tar formation, which could otherwise inhibit further reactions [43]. Hydrophobic and amphiphilic molecules may have been carried within supercritical gas bubbles, while amphiphilic molecules associated with hydrophilic compounds could have been transported upwards, in a process resembling flotation. Heat flows in microfractures of tectonic faults can contribute to the separation of prebiotically relevant molecules, thereby purifying mixtures and concentrating substances [44,45].

Pure CO_2_, with a critical point of 30.98 °C and 73.77 bar [46], enters a supercritical state (scCO_2_) at depths of around 1000 m, depending on temperature and the density of gas bubbles in the overlying water column. Similarly, pure N_2_, with a critical point of −146.9 °C and 33.96 bar [46], achieves supercriticality at approximately 400 m in an open water column. Mixtures of CO_2_ and N_2_ may exhibit supercritical fluid properties with intermediate critical points depending on their respective proportions. This behavior can also be observed for trace gases such as NH_3_, CH_4_, and SO_2_, provided they are present in sufficient concentrations and under appropriate conditions.

Under these conditions, supercritical gases function as non-polar solvents, enabling the accumulation and interaction of organic molecules. Cavities where scCO_2_/scN_2_ fills the upper region and water occupies the lower region act as autoclave-like reaction chambers with distinct two-phase boundaries. In such environments, amino acids can condense into peptides, releasing water as a by-product. This process is enhanced by pressure drops during cyclic geyser eruptions, which trigger local phase transitions from scCO_2_ to subcritical CO_2_ (gCO_2_). The accompanying entropy increase during these phase transitions facilitates peptide chain elongation without requiring catalysts [25]. Organic molecules transported from deeper crustal regions via supercritical gas bubbles would likely accumulate here due to reduced solubility in subcritical gases.

## 3. The First RNA: A Precursor to tRNA?

The formation of RNA under hydrothermal conditions remains unproven, but thermodynamic calculations suggest that RNA building blocks—adenine, guanine, cytosine, thymine, uracil, ribose, and deoxyribose—could synthesize from precursors such as CH_2_O and HCN under the temperature, pressure, and fluid composition characteristic of hydrothermal systems [34]. Given that these RNA components could form within the upper continental crust, the basic prerequisites for an RNA-based information storage system may have existed. However, as a random product of chemical reactions, RNA would not initially have served as an information storage system akin to modern mRNA or DNA. Instead, early RNA may have had an autocatalytic function, acting as a ribozyme [47,48,49].

RNA takes on a unique role if viewed as a precursor to tRNA rather than as a direct informational molecule like mRNA. RNA is most stable in acidic conditions, such as those found in low-temperature hydrothermal waters within tectonic faults [37,50]. Under these conditions, RNA remains protonated at low pH levels. However, when scN_2_ dominates the gas phase, protonation is avoided, allowing the formation of longer RNA strands. Even so, RNA strands tend to form duplexes with complementary sequences, hindering replication, as duplex structures prevent the incorporation of free nucleotides. Single-stranded RNA would eventually transition into duplexes, rendering them non-replicable and effectively forming a dead-end product [51].

Fault zones containing low-temperature hydrothermal water with excess N_2_ (and possibly CO_2_) offer optimal conditions for RNA replication. At depths below 400 m, temperatures on the early Earth could have exceeded 50 °C due to surface water infiltration from artesian aquifers. During geyser eruptions, the phase change from supercritical to gaseous N_2_/CO_2_ induces cooling due to gas expansion (Joule–Thomson effect). When water refills the fault and compresses the gas, temperatures rise, restoring supercritical conditions. These cycles, with peak temperatures surpassing 60 °C, could separate RNA duplexes, enabling replication to proceed with available building blocks once cooling occurs. The lower melting point of RNA compared to DNA may explain DNA’s absence during early cellular evolution. Early RNA strands likely differed significantly from their modern counterparts, potentially incorporating additional organic bases and sugars in both D- and L-enantiomeric forms.

Even if enantiomerically pure RNA formed, this alone would not guarantee the evolution of a robust information system. Longer strands often develop complementary base pairings between the first and last thirds, folding into loops. In modern tRNA, these loops form structures where three outward-facing bases act as anticodons that bind complementary nucleotides. A similar structure could have arisen during early chemical evolution, serving as a precursor to tRNA. In modern cells, tRNA translates specific amino acids into the genetic code. Proto-tRNAs may have played a comparable role. For such a function, the end opposite the anticodon would need to remain single-stranded, analogous to the acceptor arm of modern tRNA, which carries the conserved base sequence CCA.

## 4. The Formation of Vesicles

In laboratory experiments, the conditions of cold-water geyser systems were recreated within a high-pressure cell, where water, CO_2_, lipids, and amino acids were subjected to cyclic pressure fluctuations. This experimental setup allowed for the demonstration of vesicle formation and the chemical evolution of peptides [9,24]. The experiments focused on the transition zone from supercritical CO_2_ (scCO_2_) to gas at depths of around 1000 m, occasionally incorporating varying proportions of nitrogen (N_2_) at shallower depths. The process unfolds as follows: along fracture edges, numerous cavities form on fault surfaces, facilitating the accumulation of CO_2_. These cavities act as reaction chambers, with water occupying the lower section and scCO_2_ or gCO_2_ residing in the upper section.

When scCO_2_ is present, a decrease in pressure during a geyser eruption triggers a phase transition in the transition zone, producing gCO_2_, in which dissolved water condenses into mist. Experimental evidence suggests that this condensation facilitates vesicle formation, as lipids from scCO_2_, which cannot remain in the gas phase, accumulate to form a primary envelope on the surface of the mist droplets. As these droplets sink towards the water–gas interface, where lipids are also present, they undergo further coating, ultimately forming vesicles with bilayer membranes in the aqueous phase. During the pressure drop, amino acids link to form peptides, which can interact with the vesicle membranes. Mayer et al. [24] demonstrated that repeated pressure fluctuation cycles can drive chemical evolution, leading to the mutual stabilization of peptides and vesicles [9]. However, the vast diversity of potential peptide combinations poses a challenge to the formation of identical amino acid sequences, representing a key limitation of this process. Importantly, no mechanism exists to store sequence information, such as that provided by RNA or DNA.

A secondary vesicle formation process occurs in the gas phase alongside mist formation. During pressure drops in a geyser eruption, dissolved gas in the water is released, resulting in an eruption so vigorous that, similar to the opening of a champagne bottle, it induces turbulence and forms foam within the cavities. In this case, droplets containing lipids from the water–gas interface are propelled into the gas phase. The initial lipid envelope forms around these droplets, which then receive a secondary coating upon re-entering the interface. Notable distinctions exist between the two vesicle formation mechanisms. While mist droplets contain quasi-distilled water (with regard to dissolved inorganic substances) and forcibly absorb organic molecules from previously supercritical gas, “champagne droplets” contain salts from the fluid and organic molecules that are readily water-soluble. Additionally, the droplet sizes differ significantly, with mist droplets measuring a few microns in diameter and “champagne droplets” being approximately two orders of magnitude larger.

## 5. The Hypothetical Model for Storing Information

The model proposed here suggests that the development of an information storage system requires the presence of a proto-tRNA composed of only 12 nucleobases. This proto-tRNA would involve a single base pair, with a single strand featuring the nucleobases CCA at one end, and a loop at the other, where three bases rotate outward to form an anticodon (Figure 2).

During vesicle formation at depths exceeding 400 m, hydrothermally generated hydrophobic amino acids can form short peptides that integrate into the vesicle membrane, while hydrophilic amino acids remain in the aqueous phase. Glycine, as a neutral amino acid, can reside in either the water or scCO_2_/scN_2_ environments, depending on the specific physicochemical conditions. Given the hydrophobic nature of adenine, the single strand of the proposed proto-tRNA could embed itself within the membrane, allowing interaction with the membrane-associated hydrophobic amino acids (Figure 3). This arrangement would facilitate linkage at the 2′-OH position of the terminal ribose, occurring cyclically within the geyser system.

A critical question is how a specific bond between the corresponding amino acid and the anticodon’s base triplet could be formed. The base triplet, which ultimately defines the code, consists of three out of four possible bases, each exhibiting distinct hydrophobic or hydrophilic properties. As each base and its position within the triplet vary in hydrophobicity, they produce differing entropic forces due to the hydrophobic effect, which influences the acceptor arm’s penetration depth into the membrane. Thus, greater hydrophobicity within the anticodon region would drive the proto-tRNA’s tip deeper into the membrane, where it encounters the most hydrophobic amino acids, likely located in the innermost zone of the membrane. It is hypothesized that these amino acids were the initial constituents of short peptides and detached from the membrane as they bonded to the proto-tRNA. The presence of individual hydrophobic amino acids within the membrane is unlikely. However, in experiments on the chemical evolution of peptides, short hydrophobic peptides have been detected in the membrane [25].

Fine-tuned positioning of the CCA arm within the membrane could arise from various base combinations, allowing distinct hydrophobic amino acids to attach at specific membrane depths. When more hydrophilic bases are present within the anticodon, the CCA arm does not penetrate as deeply into the membrane, facilitating the attachment of hydrophilic amino acids from the water side at the 3′-OH position. If the terminal ribose aligns with the lipid heads at the inner edge of the membrane, no amino acid binds, corresponding to a stop codon.

The question arises as to why only one variant of the unpaired arm was selected during evolution for tRNA loading. All possible single-stranded compositions likely existed, including AAA, which might have enabled even better coupling in the membrane. However, AAA would create such strong binding that flexibility would be reduced, preventing fine regulation. Through an immense number of trials over long periods in a protected environment, the CCA combination ultimately proved to be optimal for fine regulation.

This model allows an important conclusion: if the hydrophobicity of the nucleobases played a role in the selection of amino acids, then there must be a systematic correlation between these two groups of molecules in the genetic code with regard to their physicochemical properties. Figure 4 illustrates this relationship, based on Jungck’s [52] findings, particularly highlighting the correlation among amino acids that can form hydrothermally. This relationship underscores the potential evolutionary role of physicochemical properties in the development of the genetic code.

## 6. Next Evolutionary Step: Repurposing tRNA as mRNA

The only double-stranded region of the proto-tRNA readily dissociates under increased temperatures or turbulence typical of eruptive cycles. Single strands formed this way could then be replicated, acting as proto-mRNA templates (Figure 5). After linking with complementary, weakly specifically loaded proto-tRNAs and the subsequent linking of amino acids, the first short peptide chains (tetrapeptides) would be formed, the sequences of which would be preserved in the RNA. Initially, many proto-tRNA templates may not have contributed directly to peptide synthesis, as they encoded amino acids that were absent in the environment. However, specific proto-mRNA strands could sporadically enable peptide formation, suggesting that certain sequences, under appropriate conditions, could yield functional peptides, marking an early step in protein evolution.

Given that proto-tRNA must be homochiral (existing in both D- and L-forms), a preferential linkage emerges between L-amino acids and D-ribose in RNA (and vice versa) [53]. Consequently, the peptides formed were also homochiral, existing in equal amounts of both enantiomers. With a maximum of 12 amino acids potentially linking specifically to corresponding proto-tRNAs, at least an equal number of tetrapeptide templates would have existed, likely many more. The fusion of two same-handedness tetrapeptides could yield chains long enough to traverse membrane thickness, potentially forming pores crucial for ion and molecule exchange and acting as early energy generators. While these membrane proteins initially appeared with both chiralities in equal frequency, the balance must have shifted at some point in favor of the configurations that prevail today. Since these peptide sequences were storable, evolutionary development shifted towards combining tetrapeptide chains or proto-mRNA templates, fostering the emergence of functional molecules with catalytic properties. If these molecules supported the synthesis of further functional molecules with the same chirality, they helped to displace the other chirality in the fight for resources. In this way, their own chirality finally prevailed. At the same time, they formed the basis for the formation of more complex peptides, so that the origin of life can be defined at this point in time.

The development of more complex molecules would occur in a modular manner. Specifically, different RNAs, each consisting of 12 nucleobases (four triplets), would randomly combine into longer units. With a sufficient supply of building blocks from the surrounding environment, they could be copied and/or function as mRNA. A particularly advantageous peptide, for example, would be one that catalyzed the linkage of 12-mer strands. Through the addition of these 12-mer strands, larger tRNAs or strands contributing to ribosome assembly could have evolved. For example, the small ribosomal subunit contains 120 nucleotides (10 strands). Since these strands could have initially functioned as mRNA, it should be possible to identify sequence remnants in enzymes that indicate an early linkage to rRNA. A modular assembly mechanism promotes rapid evolutionary development and does not seem uncommon in nature, as demonstrated by Lavdovskaia et al. (2024) in their study on the modular assembly of the small mitoribosomal subunit in human cells [54].

Through millions of years, peptides with novel catalytic functions emerged, enhancing reactions that stabilized vesicle formation. At a certain evolutionary point, biochemical selection enabled cells to replicate essential molecules using external building blocks. When a protocell reached sufficient size, flow-induced shear forces triggered division, allowing daughter cells to independently thrive, provided that the mother cell had accumulated surplus resources. This process led to a population of ancestor cells with distinct (quantitative) compositions, all operating with a unified “language and grammar” in molecular replication and interaction.

In the end, this model reveals a pathway that highlights the essential and dynamic interplay between RNA, peptides, and a membrane as a hydrophobic “compartment”—a crucial step in the transition from non-living matter to life. It differs from research approaches that focus on the stereochemical theory, the coevolution theory, or the error-minimization theory [55] by providing clear boundary conditions that are testable and can be experimentally replicated in the laboratory in the future.

Today, many researchers are exploring possible bridges between the very origins of life and modern biology. With the emergence of the first prokaryotes, all traces of early life development seem to have been lost. One potential link could be the way amino acids are attached to tRNA by synthetases. In modern biochemistry, two distinct classes of aminoacyl-tRNA synthetases catalyze the specific attachment of amino acids to tRNAs [56,57]. Class I synthetases, which attach amino acids at the 2′-OH end of the terminal ribose, are generally associated with larger, less polar amino acids, while Class II synthetases target the 3′-OH end, attaching smaller, more polar amino acids. Biochemical and bioinformatic analyses, alongside protein engineering studies, suggest that these two synthetase classes may have evolved from complementary strands of a common ancestral gene [56]. This proposition aligns with the hypothesis that modern aminoacyl-tRNA synthetases trace their origins back to early molecular evolution events, specifically to a template-driven mechanism akin to the proto-tRNA formation described here. The peptides synthesized on proto-tRNA and its complementary strand may later have been encoded in DNA using bidirectional genetic coding in a sense–antisense manner.

## 7. Summary

The ancient continental crust provided an optimal setting for the emergence of early life, offering a stable, sheltered environment that persisted over millions of years. This habitat was rich in essential resources and offered diverse conditions conducive to organic molecule formation, such as variations in pressure, temperature, and pH, alongside natural catalysts. The crust’s structure allowed for continuous substance exchange as an open system. Supercritical gases dissolved nonpolar compounds, thereby facilitating organic synthesis and polymer formation. The cyclic phase transitions induced by cold-water geysers generated entropy fluctuations, promoting chemical reactions without the need for activation energy. Though short-lived, vesicles were readily formed, enabling the spontaneous linkage of molecular chains driven by entropic forces. Temperature variations enabled RNA replication, while vesicle membranes supported interactions with organic molecules, including proto-tRNA. Acting as molecular bridges, proto-tRNAs linked specific amino acids and conveyed information, simultaneously establishing the precursors to mRNA. In summary, the development of life was fundamentally propelled by the dynamic interaction of three critical components: vesicles, peptides, and RNA. These interactions set the stage for the evolution of life, where entropic forces and cyclic environmental conditions drove the emergence of complex molecular systems. If rift zones in the continental crust are confirmed as a potential environment for the origin of life, this would open a new perspective on the search for life on other planets [58,59]. Similar conditions are likely to be common, suggesting that comparable organic chemical processes could have taken place. Even if no traces of life are found on the planet’s surface, remnants of past life may be preserved in the fluid inclusions of hydrothermal minerals. Against this background, targeted sampling of mineralized fault zones on Mars, for example, could be a promising approach.

## Figures and Tables

**Figure 1 life-15-00433-f001:**
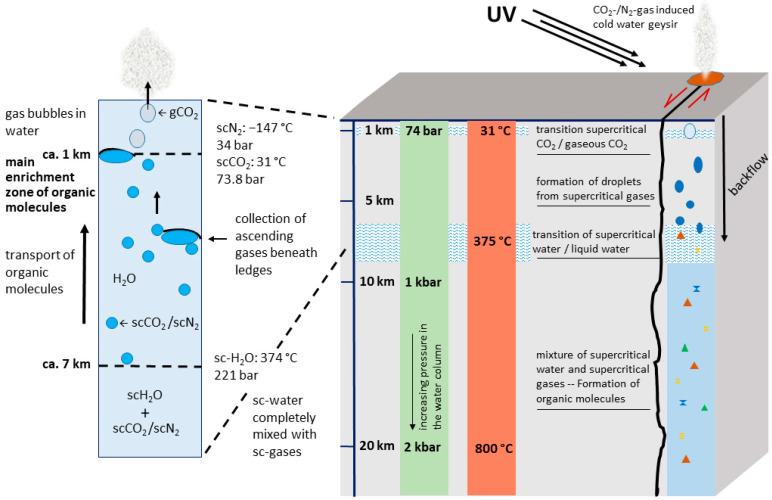
Schematic block profile of the early continental crust: The average thickness of the modern continental crust is approximately 30 km; however, during the early Earth, it was likely considerably thinner. Despite the overall higher global temperatures at the time, artesian water within the upper layers could have facilitated localized cooling, resulting in a temperature gradient within the crust. red arrows: relative movement of continental blocks.

**Figure 2 life-15-00433-f002:**
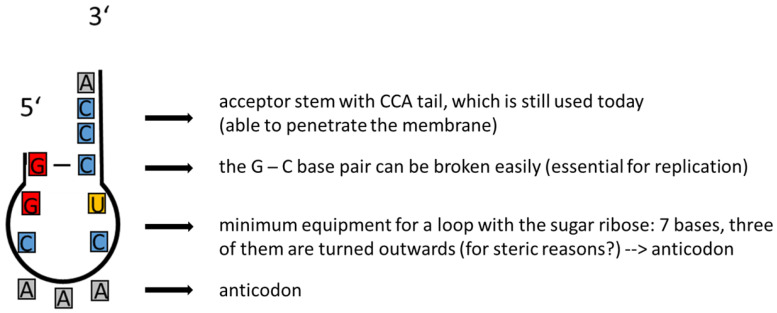
Example of a proto-tRNA consisting of only 12 nucleobases: This form of proto-tRNA would have been selected from a diverse pool of RNAs, varying in length, conformation, and molecular variations, reflecting the diversity of possible sugar molecules in both chiral forms. The selection of a specific enantiomerically pure tRNA type likely occurred over an extended period, during which the processes outlined below would have played a crucial role. The sequence would include the bases A (adenine), G (guanine), C (cytosine), and U (uracil).

**Figure 3 life-15-00433-f003:**
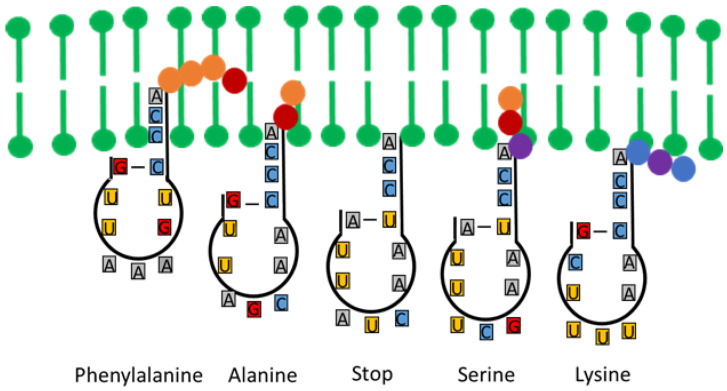
Variations in the penetration depths of the CCA acceptor arm of a proto-tRNA into a vesicle membrane (depicted in green) relative to the hydrophobicity of the anticodon. The first amino acid in each peptide sequence attaches to the CCA arm while simultaneously detaching from the peptide chain. Hydrophobic amino acids are represented by yellow and red dots, while hydrophilic amino acids are depicted as blue and purple dots. The nucleobases A (adenine), G (guanine), C (cytosine), and U (uracil) are shown.

**Figure 4 life-15-00433-f004:**
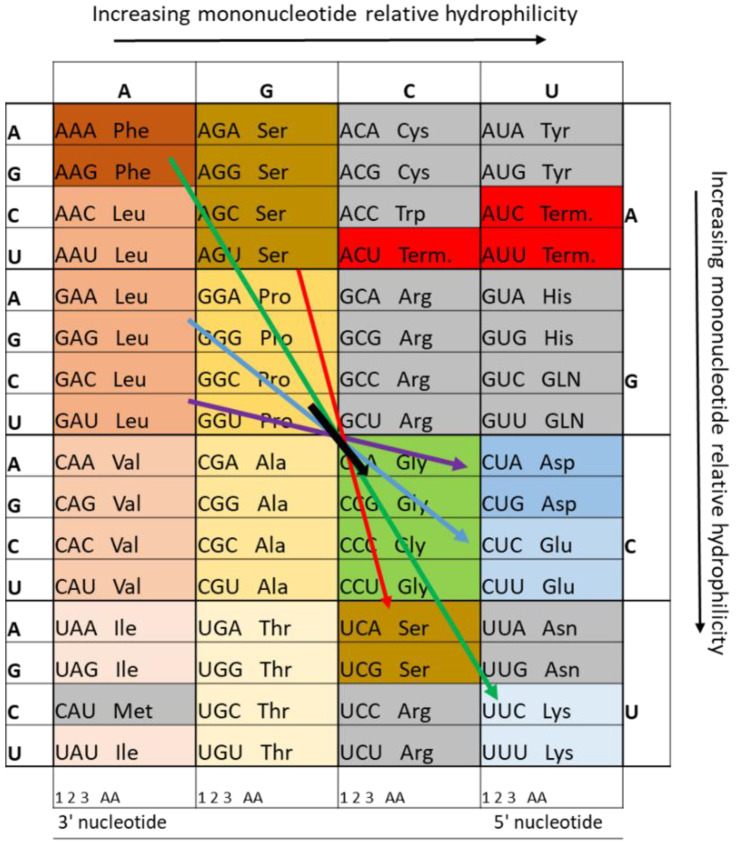
Complementary amino acid groups reflecting the utility of proto-tRNA templates and their complementary strands, following Jungck [52]. Amino acids synthesized within hydrothermal systems are highlighted in color, with stop codons indicated in red. Hydrophilic amino acids (shown in shades of blue) are positioned in the most hydrophilic anticodon region.

**Figure 5 life-15-00433-f005:**
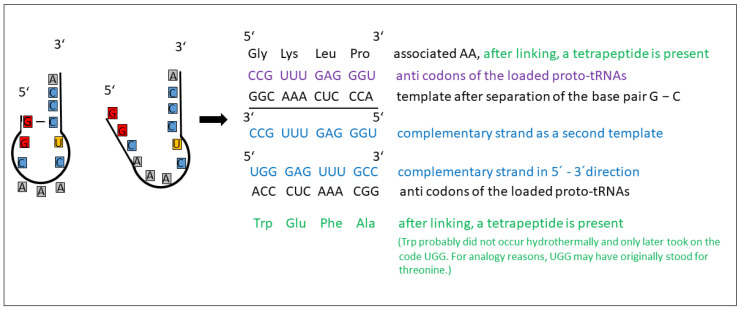
Template utilization in proto-tRNA duplication—an example illustrating the utilization of separated proto-tRNA strands as a template for peptide assembly. Once the complementary strand is synthesized, a peptide chain comprising four amino acids (shown in green) is formed along the template strand.

## Data Availability

Not Applicable.

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
