# Peer review of "The Origin of Life in the Early Continental Crust: A Comprehensive Model"

_life, 2025, doi:10.3390/life15030433_

Round 1

Reviewer 1 Report

Comments and Suggestions for Authors

There are two important parts of this paper. The first argues that fault zones in continental crust are a good site for the origin of life, and the second gives arguments for the way that proto-tRNAs were charged with amino acids and reasons for associations of particular codons with particular amino acids. Both these topics are interesting. However, I see little connection between the two. I think the author is trying to fight two battles at once, which is dangerous. Either part of the author’s argument could be true or false, independently of the other, so it might be wiser to separate them into different papers.

In particular, the proposal about charging of tRNAs (Fig. 3) requires an environment where RNA, amino acids and lipid membranes exist. It seems reasonable to propose that these things could exist on prebiotic earth, but I don’t think that limits us to fault lines in the crust. Section 3 argues for the importance of temperature cycles to separate RNA duplexes, but this could also occur in other environments. Section 4 argues that vesicles can form where there is supercritical CO2, but vesicle formation is fairly spontaneous under many conditions.

I do not feel qualified to comment on the geological aspects of this paper. I think that we are unsure of the environment in which life arose, and it is useful to consider all possible alternatives, so I am not opposed to the ideas presented here. However, I feel it is misleading to claim that the ideas about tRNAs and the genetic code depend on the geological arguments. Mixing in the geological parts makes me feel less inclined to believe the biochemical parts.

The RNA part of the paper is closer to my own expertise. Line 246 says “Given the hydrophobic nature of adenine, the single strand of the proposed proto-tRNA could embed itself within the membrane”. But what about the hydrophilic nature of ribose? Has it been shown that RNA strands embed in membranes in the way proposed? I don’t think so. And why do all tRNAs end in CCA? I think it is being argued that A is more hydrophobic than C, so if embedding in the membrane is important, why does the sequence not end in AAA?

It is argued that there is fine-tuning of the position of the CCA arm in the membrane according to the nature of the anticodon bases. This is rather difficult to believe, because the anticodon is quite far from the membrane in Fig 3, and the anticodon bases will be surrounded by water in all cases.

Fig 3 proposes several things that could in principle be tested in a lab. Do tRNAs embed in membranes? Is the depth of embedding dependent on the anticodon sequence? Does any reaction occur between the end of the RNA and peptides that may lie within the membrane? If there is a reaction, is the nature of the amino acid added dependent on the anticodon sequence? If there were some measurements of these things, this would be wonderful, but in absence of measurements I feel very doubtful about the proposed arguments.

Line 272 (caption) says “The first amino acid in each peptide sequence attaches to the CCA arm while simultaneously detaching from the peptide chain.” This is the opposite of what happens in protein synthesis, where the peptide chain has to grow. In the author’s model, why is it necessary to form peptides first and then break an amino acid from the peptide in order to bind to the tRNA? Why cannot single amino acids bind directly to the tRNA?

Reviewer 2 Report

Comments and Suggestions for Authors

Respected Author may find my few comments as follows:

1- Introduction, line 17: Please add explanations of what catalytically active peptides mean in terms of their molecular structure.

2- Page 3, line 97: It is recommended to write sulfur-based and nitrogen-based compounds with H2S and NH3 as the two basic molecules that could be formed in the early stage of Earth's geochemical environment

3-Page 4, lines 133 and 136: Please cite the references for critical point values. It is recommended that the Author cites the NIST chemistry webbook.

4- Page 4, line 137: Please re-write this sentence as "mixtures of CO2 and N2 may exhibit supercritical fluid properties".

5- Page 5, line 233: Since different bondings exist in each nucleotide, the Author needs to add a 3D rendering molecular model in addition to the schematic diagram in Figure 2, to precisely show the 3D position of the anticodon.

6- Figure 2, Typo: Please correct to anticodon.

Reviewer 3 Report

Comments and Suggestions for Authors

In this interesting work, the author presents a plausible scenario for the origin of living systems in tectonic faults. The author has done previous research in the subject, and builds on it nicely by adding some new developments. However, I recommend that the following changes are implemented before it can be reconsidered for publication.  

  1. The author is recommended to add further reviews (see below) on the origin of life in the first paragraph, because these references discuss all of the questions in more detail. In particular, there are exciting developments in computational/mathematical modeling that also merit a mention. Reviews (i.e., books and papers) on the origin of life are provided below: https://pubs.acs.org/doi/10.1021/cr2004844 https://www.cambridge.org/core/books/origin-and-nature-of-life-on-earth/CEA8B764107B896A9CBA371F552BE215 https://www.cambridge.org/core/books/emergence-of-life/16DE4E9AFAA0F4F840AC40E42C05222E https://www.hup.harvard.edu/books/9780674987579  
  2. Author writes that "black smokers" are one of the two leading theories. That is not correct, because their physical and chemical parameters are too extreme. The real candidate (also a submarine hydrothermal vent) are the alkaline hydrothermal vents at lower temperatures. These are the ones pioneered by Michael Russell, William Paul Martin, and colleagues. https://www.nature.com/articles/nrmicro1991  
  3. Work done by Dieter Braun and colleagues is lending much indirect support to the (tectonic) faults (fissures). This work should be discussed in context of the hypothesis. Two references in the area are provided: https://www.nature.com/articles/s41586-024-07193-7 https://www.nature.com/articles/s42254-022-00550-3  
  4. Acidic pH in tectonic faults can be valuable for RNA world hypothesis. This point should be further elaborated by consulting following reference: https://biologydirect.biomedcentral.com/articles/10.1186/1745-6150-7-4  
  5. There is growing evidence that atmospheric escape was crucial in perturbing early Earth, and also early Mars. Hence, this point is worth reiterating in the Conclusion, where I believe that the author can also highlight the connections with early Mars – in other words, a good site for abiogenesis on early Mars are the tectonic faults. https://iopscience.iop.org/article/10.3847/2041-8213/aac489/meta  
  6. Fig. 1 is rather hard to read even when full page. Partly because it is blurry, but partly because the text and details are too small. Please reconsider enlarging text size and removing some non-essential details for clarity.  
  7. Author mentions enantiomers and (homo)chirality in Sec. 3, but I think this point should be expanded a bit more. Are there any particular conditions that would aid emergence of homochirality in tectonic faults compared to other settings? This question is worth exploring.  
  8. With regard to supercritical CO2, the author could add a figure on the phase diagram of CO2, or at least an online reference to it, so that readers can look up the regime where it exists.  
  9.  It is interesting that just 12-nucleobase proto-RNA could have functionality. I have two questions in this context that should be discussed in the paper: (1) is there experimental support for such a small proto-RNA molecule fulfilling the necessary function(s)? (2) how would the replication of this molecule take place through some specific process?  
  10. Since the well-known work of Jungck, plenty of research has been done on amino acid selection and emergence of genetic code. I think this topic is too complicated to be properly discussed in this article. Hence, I would suggest adding some further references and perhaps 1-2 paragraphs of text. https://iubmb.onlinelibrary.wiley.com/doi/full/10.1002/iub.146 https://www.pnas.org/doi/abs/10.1073/pnas.0603780103 https://www.liebertpub.com/doi/abs/10.1089/ast.2022.0107  
  11.  With regards to Sec. 6, plenty of recent work highlighting the deep connections between peptides and RNAs: the RNA-peptide world. I think it would be helpful for the author to incorporate such results into the hypothesis.  
  12. Sec. 7 is just a single paragraph. I suggest merging with Sec. 6 into a larger section with improved flow of content.

Round 2

Reviewer 1 Report

Comments and Suggestions for Authors

The paper gives some interesting ideas and so I am willing to recommend publication.

The part about the reactions in the crust is well argued, but the authors have published similar papers previously. It is possible that it might work this way, but I don't feel convinced that it is the only solution.

The part about tRNA charging is quite novel, but not yet tested. I still do not see why tRNA charging has to happen in a membrane at all, or that tRNAs would anchor in a membrane in the way proposed. So I still think this is very speculative.

Reviewer 2 Report

Comments and Suggestions for Authors

Respected Author has addressed my comments and made the necessary corrections.

Reviewer 3 Report

Comments and Suggestions for Authors

In the revised version, the author has done a solid job of addressing and incorporating the comments from the reviewers. Moreover, the topic is interesting and timely, and the author has clearly communicated the idea(s) behind this origin-of-life proposal. Hence, I recommend that the paper is accepted for publication in its current form.